# Transcriptomic and Metabolomic Analyses of Soybean Protein Isolate on *Monascus* Pigments and Monacolin K Production

**DOI:** 10.3390/jof10070500

**Published:** 2024-07-19

**Authors:** Xueling Qin, Haolan Han, Jiayi Zhang, Bin Xie, Yufan Zhang, Jun Liu, Weiwei Dong, Yuanliang Hu, Xiang Yu, Yanli Feng

**Affiliations:** 1Hubei Key Laboratory of Edible Wild Plants Conservation and Utilization, College of Life Sciences, Hubei Normal University, Huangshi 435002, China; qxl2524157454@163.com (X.Q.); hh663503493@hbnu.edu.cn (H.H.); zhang34317778@126.com (J.Z.); xiedodo211@163.com (B.X.); 13123883302@163.com (Y.Z.); dove_lj@126.com (J.L.); weiweidong@hbnu.edu.cn (W.D.); ylhu@hbnu.edu.cn (Y.H.); xiangyu@hbnu.edu.cn (X.Y.); 2College of Life Sciences, Hubei Normal University, Huangshi 435002, China

**Keywords:** *Monascus* pigments, monacolin K, soybean protein isolate, transcriptomic, metabolomic

## Abstract

*Monascus* pigments (MPs) and monacolin K (MK) are important secondary metabolites produced by *Monascus* spp. This study aimed to investigate the effect of soybean protein isolate (SPI) on the biosynthesis of MPs and MK based on the analysis of physiological indicators, transcriptomes, and metabolomes. The results indicated that the growth, yellow MPs, and MK production of *Monascus pilosus* MS-1 were significantly enhanced by SPI, which were 8.20, 8.01, and 1.91 times higher than that of the control, respectively. The utilization of a nitrogen source, protease activity, the production and utilization of soluble protein, polypeptides, and free amino acids were also promoted by SPI. The transcriptomic analysis revealed that the genes *mokA*, *mokB*, *mokC*, *mokD*, *mokE*, *mokI*, and *mokH* which are involved in MK biosynthesis were significantly up-regulated by SPI. Moreover, the glycolysis/gluconeogenesis, pyruvate metabolism, fatty acid degradation, tricarboxylic acid (TCA) cycle, and amino acid metabolism were effectively up-regulated by SPI. The metabolomic analysis indicated that metabolisms of amino acid, lipid, pyruvate, TCA cycle, glycolysis/gluconeogenesis, starch and sucrose, and pentose phosphate pathway were significantly disturbed by SPI. Thus, MPs and MK production promoted by SPI were mainly attributed to the increased biomass, up-regulated gene expression level, and more precursors and energies.

## 1. Introduction

*Monascus* spp., as a medicinal and edible microorganism, has a long history of application in China [1]. *Monascus* spp. can produce a variety of functional secondary metabolites, mainly including *Monascus* pigments (MPs), monacolin K (MK), and γ-aminobutyric acid. Moreover, some of the *Monascus* species can also produce citrinin with nephrotoxicity [2,3]. MPs, a class of polyketide compounds, contain three colors, red, orange, and yellow, and possess abundant activities such as lipid metabolism regulation, antibacterial, weight loss, anti-inflammatory, antioxidative, and anti-tumor properties [4]. MPs are widely utilized in the food industry to substitute synthetic pigments due to their bright color, high safety, and various biological activities [2].

MK can effectively inhibit the synthesis of cholesterol and then regulate blood lipid levels. In addition, MK also shows good anti-cancer, antibacterial, and anti-inflammatory properties [5,6], which show great potential for clinical application. Moreover, MK is a key ingredient in functional red mold rice (FRMR), and it is also an index to measure the quality of FRMR [6]. The application of MPs and MK is restricted by their low yields and high costs [4,5]. In addition, the health and safety of many *Monascus*-fermented products are currently related to the concentrations of MK and citrinin [6,7,8]. Therefore, improving the production of MPs and MK and inhibiting the biosynthesis of citrinin has become an urgent problem in the *Monascus* fermentation industry.

In recent years, some strategies have been adopted to improve MPs and MK production, such as strain selection and breeding [9,10], control of fermentation conditions [11,12], optimization of medium components [2], and the addition of exogenous additives [3,13]. Among them, the optimization of nitrogen sources in the medium is one of the most commonly used ways. Several studies have indicated that the growth and secondary metabolites production of *Monascus* spp. could be significantly affected by carbon or nitrogen sources. For example, the mycelial morphology, the expression of genes related to MPs synthesis, and the production of yellow MPs could be regulated by ammonium chloride [14]. The production of hydrophilic yellow MPs in *M. ruber* CGMCC 10910 was increased by approximately 75% under the addition of nitrates [15]. Huang et al. [16] confirmed that ammonium-form nitrogen was more conducive to MPs production than nitrate and organic nitrogen. Arginine [17] and glutamate [5] affected MK production by altering cellular permeability and regulating the expression level of genes involved in MK biosynthesis. 

The experimental strain, *M. pilosus* MS-1, was isolated from commercially available red mold rice which produces both MK and MPs. The bias production of MK or MPs could be regulated by fermentation temperature [18] or additives [19,20]. For example, the MPs yield of *M. pilosus* MS-1 was 13.78 U/mL when 40 g/L glycerol was added, and the genes related to MPs biosynthesis, *ctnR* and *PKS5*, were significantly up-regulated [19]. In addition, when 60 g/L glycerol was added, the MPs production was 3.24 times that of the control group, and the genes *MpPKS5*, *mppR1*, *MpigI*, *MpFasA2*, and *MpFasB2* were significantly up-regulated [20]. 

Our previous studies indicated that MK production could be selectively promoted by soybean flour, while that of MPs was inhibited [18]. Then, we found that the key substance affecting the production of MK and MPs was soybean protein isolate (SPI) [21]. On the basis of these, the medium including SPI was optimized for bias production of MK or MPs. Although some investigations have evaluated the effects of different nitrogen sources on the production of MPs and MK, the regulatory mechanisms of SPI on the biosynthesis of MPs and MK have not been systematically investigated.

The present study aimed to investigate the regulatory mechanism of SPI on the growth, MPs, and MK production of *Monascus pilosus* MS-1. The changes in transcriptional and metabolic in response to SPI were investigated using comparative transcriptome and untargeted metabolome to identify the key genes and metabolites that either directly or indirectly regulate the MPs and MK biosynthesis pathway. It is expected to provide comprehensive insights into the regulatory mechanisms of MPs and MK biosynthesis in *Monascus* spp. by SPI.

## 2. Material and Methods

### 2.1. Strain and Culture Condition

*Monascus pilosus* MS-1 (CCTCC M 2013295) was isolated from red mold rice purchased from the local market in Wuhan, China, and it was kept in the China Center for Type Culture Collection (CCTCC, Wuhan, China). After activation, the strain was developed on potato dextrose agar (PDA) for 10 days at 30 °C. The spore suspension was adjusted to 10^6^ CFU/mL, and 10% (*v*/*v*) inoculum was incubated in a 150 mL Erlenmeyer flask containing 30 mL of synthetic medium (monosodium glutamate, 10 g/L; MgSO_4_·7H_2_O, 0.17 g/L; K_2_HPO_4_, 1 g/L; KCl, 0.5 g/L; FeSO_4_, 0.05 g/L; sucrose, 60 g/L, pH 4.0). To investigate the effect of SPI, 1% (*w*/*v*) SPI (Shanghai yuanye Bio-Technology Co., Ltd., Shanghai, China) was added into the culture medium already containing monosodium glutamate as nitrogen source (SPI group), and the medium without SPI was used as a control (CK group). After inoculation, the submerged culture of MS-1 was placed at 120 rpm at 30 °C for 2 days, and then decreased to 25 °C until 9 days [20]. 

### 2.2. Analysis of pH and Biomass

The pH value of the fermentation broth was measured by a pH meter (Shanghai PRECISION Scientific Instrument Co., PHS-3C, Shanghai, China). The mycelia were filtered, washed with distilled water, and collected. Then, the mycelia were dried at 55 °C until constant weight to determine the biomass. The biomass was expressed as mycelial dry weight obtained from one liter of fermentation broth. 

### 2.3. Analysis of MPs and MK Production

The fermentation broth was centrifuged at 8000 rpm for 10 min, and the supernatant was collected. Then, 10 mL of 75% ethanol was added to 0.3 g of mycelia and treated by ultrasound extraction for 1 h. Next, the extracts were centrifuged at 8000 rpm for 10 min and the supernatant was collected. The detection of MPs was performed at 390 nm (yellow), 460 nm (orange), and 505 nm (red) by ultraviolet spectrophotometer (UV-1900, Shimadzu, Tokyo, Japan) according to the method of Shi et al. [3] and Tong et al. [2] with minor modifications. The MPs concentration was expressed as the color value and the computational formulas are as follows: Extracellular (fermentation broth) color value (U/L)=OD × dilution factor × 1000
$$\mathrm{Intracellular}\;(\mathrm{mycelium})\;\mathrm{color}\;\mathrm{value}\;(U/L)=\frac{\mathrm{OD}\;\times \;\mathrm{dilution}\;\mathrm{factor}\;\times \;10\;\times \;\mathrm{total}\;\mathrm{biomass}\;\times \;1000}{0.3\times \mathrm{fermentation}\;\mathrm{broth}\;\mathrm{volume}}$$
Total color value (U/L)=extracellular color value+intracellular color value

The supernatant was filtered through a 0.22 μm pore-size membrane for MK detection. MK was analyzed by high-performance liquid chromatography (HPLC) (Agilent 1260, Agilent Technologies Inc., Santa Clara, CA, USA), according to the method of Feng et al. [19] with minor modifications. The HPLC analysis was performed under the following conditions: C_18_ column (Inertsil ODS-3, 4.6 mm × 250 mm, 5 μm, Shimadzu, Kyoto, Japan); pure acetonitrile: distilled water: 0.5% phosphorus acid (*v*/*v*) = 60:37:3; flow rate, 1.0 mL/min; column temperature, 25 °C; injection volume, 20 μL; detection wavelength, 238 nm.

### 2.4. Analysis of Monosodium Glutamate and Nitrogen Content in the Fermentation Broth of Monascus spp.

The analysis of monosodium glutamate was performed using an acidimeter following GB5009.43-2016 [22]. The fermentation broth was centrifugated at 8000 rpm for 10 min and the supernatant was used for the determination of monosodium glutamate. The analysis of nitrogen content was performed by Kjeldahl nitrogen determination following the method of Shi et al. [23]. The fermentation broth of *Monascus* spp. was centrifugated at 8000 rpm for 10 min and 2 mL of the supernatant was used for the determination of nitrogen content.

### 2.5. Analysis of Protease Activity

The protease activity was analyzed by ultraviolet spectrophotometry following the method described in GB/T 23527-2009 [24]. Determination of extracellular protease activity was performed as follows: 1 mL of the fermentation broth was mixed with 1 mL of sodium lactate buffer at pH 3.0, phosphate buffer at pH 7.5, and borate buffer at pH 10.5, respectively. The supernatant was collected as the crude enzyme extract after centrifugation at 8000 rpm for 10 min at 4 °C. Determination of intracellular protease activity was performed as follows: the mycelia were drained of the water and the wet weight was recorded, and then they were divided into three portions in centrifuge tubes. Then, 10 mL of acidic, neutral, and alkaline buffers were added to the three tubes, respectively. The tissue homogenizer was used for crushing for 3 s intervals every 10 s 3 times, and the supernatant was taken as a crude enzyme extract after centrifugation at 8000 rpm for 10 min at 4 °C.

### 2.6. Analysis of Soluble Protein and Polypeptide

The supernatant of the fermentation broth was collected after centrifugation at 8000 rpm for 10 min at 4 °C. The soluble protein and polypeptide were determined by the biuret method [25]. For the determination of soluble protein, 1 mL of the supernatant was added to 4 mL biuret reagent, and the mixture was set in the dark at 25 °C for 30 min. Then, the absorbance at 540 nm was determined by a UV 1900 spectrophotometer (Shimadzu Co., Kyoto, Japan). For polypeptide content, 1 mL of the supernatant was placed in 1 mL of 10% trichloroacetic acid, and the mixture was incubated at 25 °C for 30 min. Then, the mixture was centrifuged at 8000 rpm for 10 min at 4 °C. Supernatants were collected and measured according to the method of the determined soluble protein. The concentration of soluble protein or polypeptide was calculated according to the standard curve made using bovine serum albumin as the standard.

### 2.7. Analysis of Free Amino Acids (FAAs)

The fermentation broth was centrifuged at 8000 rpm for 10 min at 4 °C, and the supernatant was collected and measured according to the previous method with minor modifications [26]. The sample derivation procedure was described as follows: 100 μL of the sample solution was pipetted into an EP tube and freeze-dried, 200 μL of derivatization buffer (21.0 g NaHCO_3_ was dissolved in 470 mL distilled water and filtered, 30 mL acetonitrile was added) was added and mixed, and then 200 μL of derivatization reagent (1.0 g 2,4-dinitrofluorobenzene was dissolved in acetonitrile and the volume was adjusted to 100 mL) was added and mixed. The sample was derivatized in a 60 °C water bath for 30 min and then placed at 25 °C. Finally, 1600 μL of equilibration buffer (a mixture that included 3.4 g KH_2_PO_4_ and 145.5 mL of 0.l mol/L NaOH and then diluted to 500 mL with distilled water, pH 7.0) was added and mixed well. After derivatization, the sample was passed through a 0.22 μm pore-size filter for HPLC analysis. The HPLC analysis was performed under the following conditions: C_18_ column (SinoChrom ODS-BP, 4.6 mm × 250 mm, 5 μm, Dalian, China), mobile phase, 0.05 mol/L sodium acetate (pH 6.4), pure acetonitrile, distilled water; flow rate, 1.0 mL/min; column temperature, 25 °C; injection volume, 10 μL; detection wavelength, 360 nm. The HPLC gradient elution program is shown in Appendix A. The content of FAAs in fermentation broth was calculated from the following equation:$$X=\frac{\mathrm{Pi}\times \mathrm{Cs}\times V\times \mathrm{Vs}}{\mathrm{Ps}\times \mathrm{Vi}}$$
where X is the concentration of amino acid in the sample, mg/mL; Pi is the peak area of sample solution; Cs is the concentration of standard solution, mg/mL; V is the dilution factor of sample solution; vs. is the injection volume of standard solution, μL; Ps is the peak area of standard solution; and Vi is the injection volume of sample solution, μL. 

### 2.8. Comparative Transcriptomic Analysis

The fresh mycelia were collected after fermentation for 9 days, washed with aseptic water, and then immediately frozen in liquid nitrogen and stored at −80 °C until the RNA extraction. Total RNA was isolated by the Trizol reagent (Life Technologies, Carlsberg, CA, USA) according to the manufacturer’s protocol. The RNA concentration and integrity were measured following the method described by Shi et al. [20], and sequencing libraries were constructed using Hieff NGS Ultima Dual-mode mRNA Library Prep Kit for Illumina (Yeasen Biotechnology (Shanghai) Co., Ltd., Shanghai, China) at Biomarker Technologies Co., (Beijing, China). Raw sequences were transformed into clean reads after data processing. These clean reads were then mapped to the reference genome *Monascus ruber* NRRL1597-16 by Hisat2. Fragments per kilobase of transcript per million fragments mapped (FPKM) were used to evaluate the level of gene expression. Each group of samples used for RNA-seq data analysis was set with three biological replicates. The RNA-seq raw data have been deposited in the NCBI database (ID: PRJNA1107712). 

DESeq2 was used to analyze the differential expression of samples. Benjamini and Hochberg’s approach was used to correct the *p*-value and the false discovery rate (FDR) was obtained. The fold change (FC) ≥ 2 and FDR < 0.01 were used as screening criteria for identifying differentially expressed genes (DEGs) [2]. The clusterProfiler packages-based Wallenius non-central hypergeometric distribution was used for gene ontology (GO) enrichment analysis of the DEGs’ functions. Functional annotation and metabolic pathway enrichment analysis of genes were carried out by the Kyoto Encyclopedia of Genes and Genomes (KEGG) database.

### 2.9. Real-Time Quantitative PCR (RT-qPCR) Validation of RNA-Seq Data

Gene expression was verified by RT-qPCR according to the methods of Shi [20]. Total RNA was extracted from mycelia using the Trizol reagent (Life Technologies, CA, USA), and cDNA was synthesized using SynScript®Ⅲ RT SuperMix for qPCR (TSK314S). RT-qPCR was accomplished by the QuantStudio stepone Plus (ABI, Los Angeles, CA, USA). The *actin* gene was used as the reference gene. The relative gene expression levels were calculated using the 2^−△△Ct^ method [20]. Appendix A shows the RT-qPCR primer information.

### 2.10. Untargeted Metabolomic Analysis

The fermentation broth and mycelia of *Monascus* spp. were collected and isolated after fermentation for 9 days. The fermentation broth was centrifuged at 8000 rpm for 10 min at 4 °C, and the supernatant was collected. The mycelia were washed with precooled physiological saline and collected [1]. Then, the fermentation broth and mycelia were frozen immediately in liquid nitrogen and stored at −80 °C until analysis [1]. Three biological replicates were used for untargeted metabolomic analysis.

For metabolite extraction, 500 μL of extraction solution (methanol:acetonitrile = 1:1, *v*/*v*) containing internal standard was added to 100 μL of fermentation broth, vortexed for 30 s, and then sonicated in an ice-water bath for 10 min. Then, 1000 μL of extraction solution (methanol/acetonitrile/water = 2:2:1, *v*/*v*) containing internal standard was added to 50 mg of mycelia, vortexed for 30 s, with a steel ball added, ground at 45 Hz for 10 min, and ultrasonicated for 10 min in ice water bath. The internal standard concentration was 20 mg/L. Then, the fermentation broth and mycelia samples were maintained at −20 °C for 1 h. The samples were centrifuged at 12,000 rpm for 15 min at 4 °C. After, 500 μL of the supernatant was dried in a vacuum concentrator, and then 160 μL of extraction solution (acetonitrile/water 1:1, *v*/*v*) was used to reconstitute it, before being vortexed for 30 s and sonicated in the ice-water bath for 10 min. We centrifuged the samples at 12,000 rpm for 15 min at 4 °C, and then took 10 μL of each sample and mixed them into the QC samples for detection.

The metabolomics analysis was composed of Waters Acquity I-Class PLUS ultra-high performance liquid tandem Waters Xevo G2-XS QT of a high-resolution mass spectrometer (Waters Co., Waltham, MA, USA). The metabolites were separated by Waters Acquity UPLC HSS T3 column (2.1 mm × 100 mm, 1.8 μm) (Waters Co., Waltham, MA, USA); the mobile phase was 0.1% formic acid aqueous solution and 0.1% formic acid acetonitrile with a flow rate of 400 μL/min and the injection volume was 1 μL. The gradient program is shown in Appendix A. 

The primary and secondary mass spectrometry data were collected and the dual-channel data acquisition mode was adopted. The low and high collision energies were 2 V and 10–40 V respectively, and the scanning frequency was 0.2 s for a mass spectrum. The parameters of the ESI ion source were listed as follows: capillary voltage, 2000 V or −1500 V in positive and negative modes, respectively; cone voltage, 30 V; ion source temperature, 150 °C; desolvent gas temperature, 500 °C; backflush gas flow rate, 50 L/h; gesolventizing gas flow rate, 800 L/h. Progenesis QI software v2.0 was used to process metabolite data, based on Biomark’s self-built library for identification. Principal component analysis (PCA) was used to evaluate the repeatability and differences in samples. Orthogonal projections to latent structures discriminant analysis (OPLS-DA) was used in multivariate analysis of the two groups, and the reliability of the model was verified. A *t*-test was used to calculate the significant difference *p*-value of each compound, and the variable importance in the projection (VIP) value of the model was calculated using multiple cross-validations. Metabolites with fold change ≥ 1, VIP ≥ 1, and *p*-value < 0.05 were defined as differential metabolites. The KEGG database was used for functional annotation and metabolic pathway enrichment analysis of differential metabolites.

### 2.11. Statistical Analysis

Three parallel replicates were set in all experiments, and the results are expressed as the mean ± standard deviation. Statistical differences between the two groups were determined by *t*-test in SPSS 27 software (IBM, Armonk, NY, USA), and *p* < 0.05 and *p* < 0.01 were considered statistically significant. Graphic generation used Origin 2022 (OriginLab, Northampton, MA, USA) and GraphPad Prism 9.5 (GraphPad Inc., San Diego, CA, USA).

## 3. Results

### 3.1. Effects of SPI on the Growth, MPs, and MK Production of Monascus spp.

The effect of SPI on the growth, MPs, and MK production of *M. pilosus* MS-1 was investigated, and the results are shown in Figure 1. On the 9th day of fermentation, the pH values in the CK and SPI groups were close to neutral (Figure 1A). The biomass of *Monascus* spp. in SPI-supplemented medium reached 22.79 g/L, representing an 8.20 fold increase over that of CK (2.78 g/L) (Figure 1B). The biomass of *Monascus* spp. was significantly enhanced by SPI (*p* < 0.05). This might be attributed to the protease decomposing SPI into smaller soluble proteins, peptides, and amino acids [25]. These nutrients were beneficial to the growth of *Monascus* spp.

The yields of red, orange, and yellow MPs in SPI-treated samples reached 398.80 U/L, 582.63 U/L, and 4035.16 U/L, respectively, which were 0.91, 1.43, and 8.01 times higher than those of CK (Figure 1C). The results showed that the production of MPs promoted by SPI, especially yellow MPs, was consistent with our previous research results [27]. Yin et al. [28] showed that the production of MPs could be promoted by histidine, lysine, tryptophan, tyrosine, and phenylalanine. It is speculated that the increase in MPs production may be related to the utilization of amino acids which were produced by SPI’s decomposition.

The MK yield of 108.61 mg/L was obtained from the SPI-treated sample, which was 1.91 times higher than that of CK (56.83 mg/L) (Figure 1D). This was consistent with the results of Shi et al. [21]. Moreover, studies have shown that MK production could be promoted by arginine [17], glutamate [29], flavin mononucleotide, arginine, lysine, and phenylalanine [30]. In addition, soybean substrate could be decomposed into small molecular proteins, peptides, and amino acids by microbial fermentation, thus affecting the growth and metabolism of microorganisms [12]. We hypothesized that the increase in MK yield was closely related to the utilization of small molecular proteins, peptides, and amino acids.

### 3.2. Analysis of Monosodium Glutamate, Nitrogen Content, Protease Activity, Soluble Protein, Polypeptides, and Free Amino Acids

As shown in Figure 2A,B, the monosodium glutamate content and nitrogen content were detected after 9 days of fermentation. A significant reduction in monosodium glutamate content was observed in the fermentation broth treated with SPI, while the nitrogen content was significantly increased (*p* < 0.05), indicating that SPI led to the utilization of monosodium glutamate by *Monascus* spp. and more nitrogen sources were provided for the growth of *Monascus* spp.

As shown in Figure 2, the effect of SPI on protease activity was determined. In the SPI-treated samples, the acid protease and neutral protease activities reached 1270.37 U/L and 12,691.33 U/L, respectively, which were 2.70 and 21.88 times higher than those of CK, respectively, indicating that the activities of acid and neutral protease were significantly enhanced by SPI, especially for that of neutral protease. However, the alkaline protease activity was extremely low during *Monascus* fermentation and could not be detected. The results of protease activity were consistent with the biomass of *Monascus* spp., suggesting that the increase in protease activity was closely related to mycelia growth and reproduction. 

The concentrations of soluble proteins and polypeptides were also evaluated under the action of SPI (Figure 2D). In the SPI-treated samples, the concentrations of soluble proteins and polypeptides reached 3.44 mg/mL and 0.47 mg/mL, respectively, which were 3.34 and 2.24 times higher than that of CK, indicating that the production of soluble proteins and polypeptides was significantly promoted by SPI. This might be attributed to the fact that *Monscus* spp. grew rapidly in the early fermentation period and a large number of proteases were secreted to accelerate proteolysis; thus, the soluble proteins and polypeptides were produced and used for growth and metabolite synthesis by *Monasucs* spp. [12]. In addition, combined with the increase in biomass, it was speculated that the increase in soluble proteins and polypeptides might be attributed to the increase in living cells and the enhancement in metabolic activity, which was similar to the results of the soybean meal fermented by *M. purpureus* 04093 [25].

The FAAs’ analysis results are shown in Appendix A. The variety of FAAs increased, while total amino acid content decreased on the 9th day of *Monascus* fermentation, indicating that the production and utilization efficiency of FAAs were promoted by SPI. The results of FAAs were consistent with soluble protein, polypeptide, and protease activity. The above results indicated that the secretion of a large number of proteases was promoted by the proliferation and metabolic activities of *Monascus* spp. to decompose SPI, resulting in an increased content of soluble proteins, polypeptides, and FAAs. In addition, the utilization of these nutrients led to the growth of *Monascus* spp. and the synthesis of MPs and MK.

### 3.3. Analysis of the Transcriptomic Data

To explore the potential regulatory mechanisms of SPI affecting the production of MPs and MK by *M. pilosus*, transcriptome sequencing was analyzed based on RNA-seq technology. The total number of clean data was 39.82 Gb, and each sample reached 6.21 Gb. The content of GC was more than 51%, and both Q20 and Q30 were more than 92%, indicating that the sample sequencing purity was high and the transcriptome sequencing data were qualified (Appendix A). The comparison rate of reads and reference genome sequences ranged from 94.28% to 96.38%, indicating the validity of the transcriptome sequencing data [31]. Pearson’s correlation coefficient indicated that the samples from the intragroup with high similarity and the data quality control were sufficient (Appendix A). At the same time, principal component analysis (PCA) showed that samples from the intragroup showed good aggregation, while the intergroup of samples showed an obvious tendency to separate in the first principal component (PC1) (Appendix A). 

Transcriptome analysis was performed to evaluate the genetic changes in *M. pilosus* under SPI treatment. The fold change and *p*-value of DEGs were displayed by volcano plots (Figure 3A). A total of 6686 DEGs were detected between the CK and SPI groups, including 3316 down-regulated genes and 3370 up-regulated genes. To intuitively display the expression differences in genes between different groups, genes with the same or similar expression patterns in different samples were clustered [2]. The hierarchical clustering analysis of the screened DEGs is shown in Figure 3B. These results indicated that SPI had a significant effect on the gene expression of *M. pilosus*.

All genes obtained by transcriptome assembly were annotated by eight functional databases, including COG, GO, KEGG, KOG, NR, Pfam, Swiss-Prot, and eggnog [2], for functional annotation and statistics of DEGs (Appendix A). To further evaluate the results of RNA-seq data, the DEGs were classified and counted using the COG database (Figure 3C). The DEGs annotated in the COG database were divided into 26 functional categories, in which carbohydrate transport and metabolism were the most annotated functional categories (275), followed by general function prediction only (202) and amino acid transport and metabolism (180), secondary metabolites biosynthesis, transport and catabolism (129), and lipid transport and metabolism (128). GO enrichment analysis showed that the ontology was involved in 35 metabolic pathways and was mainly enriched in biological processes (19), followed by molecular functions (13) and cellular components (3) (Figure 3D). The specific secondary metabolic pathways related to MPs and MK biosynthesis needed to be further evaluated by KEGG enrichment analysis. The DEGs were annotated and classified in the KEGG database (Figure 3E), and then the top 20 pathways were screened (Figure 3F). KEGG enrichment analysis showed that there were 123 KEGG pathways, and the metabolism had the most annotation information (68.48%), indicating that the metabolism of *M. pilosus* was promoted by SPI. The KEGG bubble diagram showed that the top 20 enrichment pathways included glycolysis/gluconeogenesis, pyruvate metabolism, carbon metabolism, porphyrin and chlorophyll metabolism, ascorbate and aldarate metabolism, tryptophan metabolism, steroid biosynthesis, starch and sucrose metabolism, fatty acid degradation, amino sugar and nucleotide sugar metabolism, methane metabolism, tricarboxylic acid (TCA) cycle, and amino acid metabolism. It involved arginine and proline metabolism, phenylalanine metabolism, valine, leucine and isoleucine degradation, histidine metabolism, beta-alanine metabolism, and other metabolic pathways. These results suggested that SPI significantly promoted the metabolic network of *M. pilosus* and regulated the metabolic pathways of glycolysis/gluconeogenesis, pyruvate metabolism, carbon metabolism, fatty acid degradation, TCA cycle, and amino acid metabolism, which provided abundant precursors for the formation and accumulation of secondary metabolites. 

### 3.4. Effect of SPI on the Expression of Genes Related to MK Biosynthesis

The effects of SPI on the expression of genes related to MK biosynthesis were analyzed (Figure 4). According to the results of NR and Swiss-Prot database, seven genes related to MK biosynthesis, *mokA*, *mokB*, *mokC*, *mokD*, *mokE*, *mokI*, and *mokH*, were significantly up-regulated by SPI (Figure 4A,B). These results were consistent with the yield of MK, indicating that SPI could promote the production of MK by regulating the expression of genes related to MK biosynthesis. The same conclusion was obtained by analyzing the expression profile of genes related to MK biosynthesis under the addition of SPI (Figure 4C). Among these genes, *mokA* and *mokB* encode polyketide synthase, *mokC* encodes P450 monooxygenase, *mokD* encodes oxidoreductase, *mokE* encodes dehydrogenase, *mokH* acts as transcription factor, and *mokI* acts as an efflux pump [29].

Previous studies confirmed that the production of MK was affected by the regulation of related genes. Zhang et al. [32] showed that the increase in MK production could be promoted by the overexpression of genes *mokC*, *mokD*, *mokE*, and *mokI*. Lin et al. [33] also confirmed that the expression of gene *mokE* was positively correlated with the production of MK. In addition, the overexpression of gene *mokH* significantly promoted the production of MK [34]. The gene *mokI* acts as an efflux pump, transferring intracellular MK to extracellular, promoting the final MK content [29]. Based on the above results, we hypothesized that SPI influenced the production of MK by up-regulating the expression of genes involved in the biosynthesis of MK in *M. pilosus*.

Ten DEGs were selected for RT-qPCR verification of RNA-seq data (Figure 4D). These ten genes include gene-315043, gene-213838, gene-256816 (glycolysis/gluconeogenesis, hexokinase A), gene-52089 (glycolysis/gluconeogenesis, phosphoglycerate kinase), gene-170694, gene-272292 (glycine, serine and threonine metabolism, serine hydroxymethyltransferase), gene-99680 (alanine, aspartate and glutamate metabolism, glutamate-ammonia ligase), gene-87025 (cysteine and methionine metabolism, homocysteine methyltransferase), gene-304829 and gene-194396 (glycolysis/gluconeogenesis, hypothetical protein MPDQ-000002). The result confirmed that the expression of ten tested DEGs generally matched the RNA-seq results, indicating the validity of the RNA-seq results. 

### 3.5. Analysis of the Metabolomic Response

We have confirmed that SPI regulated the gene expression of *Monascus* spp. to affect the production of MPs and MK. Metabolome was used to analyze the key metabolites and metabolic pathways that affected the production of MPs and MK. The metabolites of fermentation broth (extracellular) and mycelia (intracellular) were detected on the 9th day of fermentation by LC-QTOF. 

The results of PCA showed a significant difference in the metabolites between the CK and SPI groups, as well as that of intracellular and extracellular metabolites (Appendix A). As shown in Appendix A, the R^2^X, R^2^Y, and Q^2^Y values were all more than 0.8 or even close to 1, indicating that the predictability and stability of the OPLS-DA model were sufficient. Therefore, differential metabolites could be screened according to VIP analysis. The hierarchical cluster analysis showed that the metabolites between the CK and SPI groups, as well as that of intracellular and extracellular metabolites, were also separated obviously (Figure 5A,B).

Significant differential metabolites were screened according to the standard of fold change ≥ 1, VIP ≥ 1, and *p*-value < 0.05. The results of the statistical table of differential metabolites are shown in Appendix A. A total of 2731 differential metabolites were identified, including 1449 intracellular differential metabolites, of which 969 were significantly up-regulated and 480 were down-regulated compared to that of CK. There were 1818 extracellular differential metabolites, of which 1029 were significantly up-regulated and 789 down-regulated. These results illustrated that metabolic changes occurred under the addition of SPI. We present a visual analysis of differential metabolites in positive and negative modes using volcano plots in Figure 5C–F. The top 15 most significant differential metabolites between different groups in positive and negative modes are listed in Table 1 and Table 2, respectively. These differential metabolites were mainly involved in lipid metabolism, amino acid metabolism, carbon metabolism, sugar metabolism, and nucleotide metabolism.

The enrichment of significant differential metabolites was analyzed by KEGG (Figure 6A). The top 20 entries with the most annotated differential metabolites in the pathway were selected, and the intracellular and extracellular annotation results are shown in Figure 6B and Figure 6C, respectively. The main metabolic pathways enriched in intracellular and extracellular of *Monascus* spp. include amino acid metabolism, sugar metabolism, lipid metabolism, porphyrin metabolism, etc. Amino acid metabolism was the main metabolic pathway enriched in *Monascus* spp. under the treatment of SPI. Other important pathways included sugar metabolism, lipid metabolism, glutathione metabolism, pyruvate metabolism, and the TCA cycle. The results revealed that the intracellular and extracellular metabolic pathways of *Monascus* spp. were significantly affected by SPI.

## 4. Discussion

In the present study, SPI significantly promoted the growth of *M. pilosus* and the production of MPs and MK in liquid-state fermentation. It has been reported that among the six nitrogen sources named, including beef extract, peptone, soybean meal, yeast extract, ammonium sulfate, and sodium nitrate, soybean meal was the most suitable nitrogen source for MK production by *M. ruber* [35]. Our previous study also demonstrated that the production of MK by *M. pilosus* could be significantly promoted by soybean flours [36]. In addition, we monitored the utilization of nitrogen sources during the fermentation process of *Monascus* spp. and the results showed that SPI promoted the utilization of nitrogen sources and the production and utilization of FAAs, soluble protein, and polypeptides. Therefore, we speculated that SPI promoted the decomposition and utilization of nitrogen sources by *Monascus* spp., thereby promoting the growth of *Monascus* spp. and the production of secondary metabolites.

To better reveal the effect of SPI on the production of MPs and MK by *Monascus* spp., transcriptomic and metabolomic analyses were used to explore the differential gene expression and differential metabolite changes in *M. pilosus*. Transcriptome results showed that the expression of MK biosynthesis-related genes *mokA*, *mokB*, *mokC*, *mokD*, *mokE*, *mokI*, and *mokH* were significantly up-regulated in the SPI group, which was consistent with the production of MK. We hypothesized that SPI influenced the production of MK by up-regulating the expression level of genes involved in the biosynthesis of MK in *M. pilosus*. Transcriptome sequencing and metabolome results annotated by the KEGG database showed that SPI addition significantly affected the metabolism of *M. pilosus* MS-1, mainly involving the amino acid metabolism, glycolysis/gluconeogenesis, pyruvate metabolic pathway, TCA cycle, and lipid metabolism, accompanied by significant changes in plenty of related genes and metabolites. The underlying changes in the MPs and MK production by SPI is shown in Figure 7.

In the amino acid metabolism pathway, the transcript level of genes encoding enzymes such as serine hydroxymethyl transferase, aspartate aminotransferase, aspartate transaminase aat1, glutamate-ammonia ligase, proline dehydrogenase, glycine decarboxylase subunit P, tryptophan synthetase, amino methyl transferase, and glutamine-fructose-6-phosphate transaminase were significantly up-regulated. At the same time, metabolites such as leucine, lysine, phenylalanine, and histidine were significantly up-regulated. However, isoleucine, glutamic acid, methionine, valine, and aspartate were significantly down-regulated. The above results indicated that SPI could significantly change the amino acid metabolism process during *Monascus* fermentation. These results may also explain the higher consumption rate of monosodium glutamate and nitrogen source of the medium in the SPI treatment group than in the CK group.

Amino acid metabolism provides energy and substrates for the growth and metabolism of fungi. Glutamate, aspartate [37], lysine, arginine, and proline [1] can participate in the TCA cycle. The degradation of alanine [38], valine, leucine, and isoleucine can produce acetyl-CoA and malonyl-CoA [31]. In addition, amino acid metabolism produces amine derivatives that constitute MPs [11]. Some amino acids can not only form MPs and MK as precursors but also regulate the production of secondary metabolites by regulating the transcription of related genes or the expression of enzymes. Histidine enhances the transcription of genes in the PKS cluster but methionine shows inhibition effects [28]. Glutamate [29] and arginine [17] up-regulate the genes related to MK synthesis, change the morphology of *Monascus* spp. and stimulate the extracellular secretion of MK. Based on the above results, we speculated that the production of MPs and MK could be effected by amino acid metabolism directly or indirectly. First, the precursors for MPs and MK production such as acetyl-CoA and malonyl-CoA could be generated by amino acid metabolism. Second, the transcription of polyketide-related genes was up-regulated. Third, the expression of key enzymes for the production of MPs and MK was also enhanced. Moreover, some amino acids and their derivatives also directly participated in MPs synthesis.

In the carbon metabolism pathway, 92 DEGs were significantly up-regulated (71.88%), indicating that SPI significantly affected the carbon metabolism of *M. pilosus*. Carbon metabolism is the basic metabolism in *Monascus* spp., providing precursors and cofactors which are required by the synthesis of secondary metabolites. The C/N ratio of the culture medium is one of the important factors affecting the growth and metabolism of *Monascus* spp. [2]. Glycolysis/gluconeogenesis was the most prominent in central carbon metabolism and other important pathways including pyruvate metabolism, TCA cycle, starch and sucrose metabolism, and the pentose phosphate pathway.

In glycolysis/gluconeogenesis, the genes encoding enzymes such as pyruvate kinase, phosphoglycerate kinase, triosephosphate isomerase, glucose-6-phosphate isomerase, glyceraldehyde-3-phosphate dehydrogenase, phosphoglucomutase-3, hexokinase A, fructose-bisphosphate aldolase, and pyruvate decarboxylase were significantly up-regulated in the SPI group, which led to more energy and substrate production. Correspondingly, metabolites such as salicin, pyruvic acid, dihydroxyacetone phosphate, D-glyceraldehyde 3-phosphate, 3-ketosucrose, uridine diphosphate glucose, and deoxyribose were significantly up-regulated, while 3-phosphoglycerate, D-fructose, sucrose, and cellobiose were significantly down-regulated in sugar metabolism including glycolysis/gluconeogenesis, starch and sucrose metabolism, and the pentose phosphate pathway. The conversion of phosphoenolpyruvate to pyruvate is catalyzed by pyruvate kinase. The glycolysis pathway promotes the synthesis of pyruvate, which is converted into acetyl-CoA and flows to the TCA cycle. Acetyl-CoA is a key substance in the synthesis of polyketides and can be used as a precursor for the synthesis of MPs, MK, and fatty acids [38,39]. Polysaccharides are hydrolyzed into monosaccharides in organisms and then go through a series of pathways such as glycolysis/gluconeogenesis, the pentose phosphate pathway, and the TCA cycle to produce energy and precursors. The glycolytic intermediates are used by the pentose phosphate pathway to produce pentose sugars, which are required for the biosynthesis of nucleotides and aromatic amino acids [40]. Thus, SPI might lead to more energy and substrate production, which was beneficial for the growth of *Monascus* spp. and the synthesis of MPs and MK.

In the pyruvate metabolism pathway, the related genes encoding malate dehydrogenase, protein kinase C-like 1, fumarase fum1, pyruvate kinase, acetyl-CoA hydrolase, dihydrolipoamide acetyltransferase component pyruvate dehydrogenase complex, and probable L-lactate dehydrogenase were significantly up-regulated. However, genes encoding pyruvate carboxylase and the alpha subunit of pyruvate dehydrogenase were down-regulated. At the same time, metabolites such as pyruvaldehyde, fumaric acid, D-lactic acid, pyruvate, and L-malic acid were also significantly up-regulated, while oxoglutaric acid was significantly down-regulated. These results indicated that SPI significantly affected pyruvate metabolism. Pyruvate decarboxylase is a key enzyme that catalyzes the formation of acetyl-CoA and then polymerization to malonyl-CoA [41]. Dihydrolipoamide acetyltransferase component pyruvate dehydrogenase complex catalyzes the conversion of pyruvate to acetyl-CoA, which promotes MPs production [42]. Pyruvate can be converted to oxaloacetic acid by pyruvate decarboxylase to participate in the TCA cycle and can also be oxidized to acetyl-CoA, which is the precursor of MPs and MK [38]. In summary, we speculated that pyruvate metabolism of *Monascus* spp. could be promoted by SPI, which enhanced the production of acetyl-CoA, fumaric acid, and malic acid, thus improving the production of MPs and MK.

In the TCA cycle, the transcriptional levels of genes encoding malate dehydrogenase, aconitate hydratase mitochondrial, fumarase fum1, succinate dehydrogenase flavoprotein subunit, 2-oxoglutarate dehydrogenase E1 component, succinate dehydrogenase complex, isocitrate dehydrogenase, and citrate synthase were up-regulated by SPI. Correspondingly, metabolites such as fumaric acid and L-malic acid were also significantly up-regulated, while oxoglutaric acid was significantly down-regulated. Thus, SPI generated extensive metabolic perturbations, resulting in alterations in the transcription levels and metabolic pathways of *M. pilosus*. The TCA cycle was changed by SPI, which affected the production of MPs and MK, which was consistent with the results obtained by Zhang [43]. The metabolic intermediates of the TCA cycle, fumarate and malate, were beneficial to the production of red MPs and the transformation from orange MPs to yellow MPs [43]. In addition, the production of MK could be promoted by fumarate, malate, and citrate [5]. In short, the TCA cycle was the core pathway of energy and material metabolism in organisms, and organic acids such as fumarate and malate produced by the TCA cycle provided energy and precursors for the production of MPs and MK.

For lipid metabolism, the transcription levels of 3-ketoacyl-CoA thiolase, acetyl-CoA acetyltransferase, putative acyl-CoA dehydrogenase, and formate dehydrogenase (NAD^+^) in the fatty acid degradation pathway were significantly changed by the SPI, and the metabolites in arachidonic acid metabolism, unsaturated fatty acid biosynthesis, and linoleic acid metabolism were also significantly changed. The significantly up-regulated differential metabolites mainly included tetracosenoyl-CoA, montanoyl-CoA, docosenoyl-CoA, hexacosanoyl-CoA, linoleic acid, cerotoyl-CoA, and lignoceric acid. The protective mechanism of cell membranes and the proportion of unsaturated/saturated fatty acids affect the fluidity and permeability of cell membranes, which can be influenced by fatty acids [44]. Our previous experiments have shown that SPI promoted the secretion of extracellular MPs and MK [27]. It was speculated that the fluidity and permeability of cell membranes were regulated by SPI, which affected the secretion of MPs and MK from intracellular to extracellular. In addition, the synthesis and accumulation of MPs and MK were affected by the synthesis and metabolism of fatty acids. Caprylic acid and capric acid are precursors of MPs biosynthesis, and increasing fatty acid content provides more precursors for the synthesis of MPs [38]. Linoleic acid can increase cAMP concentration and activate the PKA pathway, up-regulate the expression of genes related to MK biosynthesis, and promote the growth of *Monascus* spp. and the production of MPs and MK [3,45]. Fatty acids such as capric acid and caprylic acid are precursors of MPs synthesis [38]. Based on these results, we speculate that lipid metabolism was changed by SPI, which affected the growth of *Monascus* spp. and the production and secretion of secondary metabolites.

Other metabolic pathways such as porphyrin and chlorophyll metabolism, ascorbate and aldarate metabolism, steroid biosynthesis, amino sugar and nucleotide sugar metabolism, glutathione metabolism, the metabolism of terpenoids and polyketides, membrane transport, nucleotide metabolism, and cofactors and vitamins metabolism were also significantly changed by SPI, indicating that the regulation of *Monascus* spp. metabolism by SPI was the result of multiple pathways. The metabolome results were consistent with the results of the transcriptome, basically indicating that changes in gene expression level and metabolites cooperatively effected the metabolic network of *Monascus* spp. in response to SPI treatment. In summary, the underlying effect behind the enhancement of the MPs and MK production by SPI was the promotion of enzyme expression levels and metabolites that participate in amino acid metabolism, carbon metabolism (glycolysis/gluconeogenesis, starch and sucrose metabolism, pentose phosphate pathway, pyruvate metabolism, TCA cycle), and lipid metabolism to provide energy and accelerate precursor accumulation.

## 5. Conclusions

In this study, the effects of SPI on the growth, gene transcription, and secondary metabolism of *M. pilosus* in liquid-state fermentation were studied based on transcriptomics combined with metabolomics. The results indicated that the gene expression and metabolite changes in *Monascus* spp. involved in amino acid metabolism, pyruvate metabolism, TCA cycle, lipid metabolism, glycolysis/gluconeogenesis, starch and sucrose metabolism, and the pentose phosphate pathway during *Monascus* fermentation were significantly affected by SPI. In other words, as a nitrogen source, SPI caused a series of metabolic changes in *Monascus* spp. at the transcriptional and metabolic levels, which affected the growth, MPs, and MK production of *Monascus* spp. In the future, the molecular mechanism of SPI on the synthesis of MPs and MK can be further studied. This work provided insights into the effect of MPs and MK synthesis in *Monascus* spp. and is expected to lay a foundation for rationally enhancing the production of MPs and MK.

## Figures and Tables

**Figure 1 jof-10-00500-f001:**
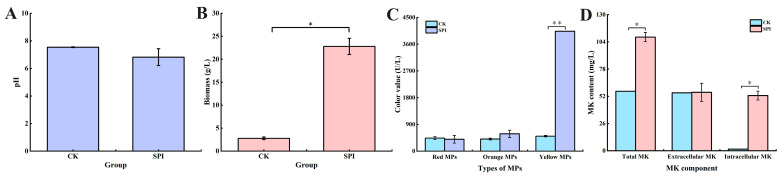
Effect of SPI on the growth, MPs, and MK production of *M. pilosus.* (**A**) pH of the fermentation broth; (**B**) biomass; (**C**) color value; (**D**) MK content. * *p* < 0.05, ** *p* < 0.01.

**Figure 2 jof-10-00500-f002:**
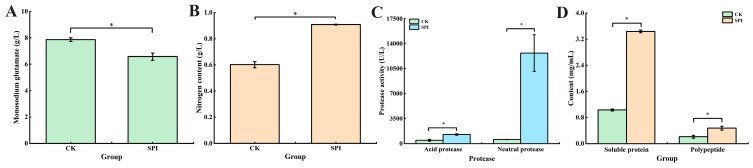
Effect of SPI on the process of nitrogen utilization by *M. pilosus.* (**A**) Monosodium glutamate; (**B**) nitrogen content; (**C**) protease activity; (**D**) soluble protein and polypeptide. * *p* < 0.05.

**Figure 3 jof-10-00500-f003:**
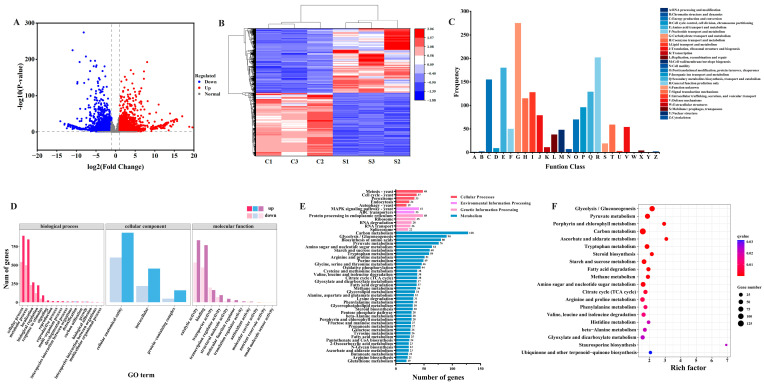
Transcriptional analysis of *M. pilosus* MS-1 response to SPI. (**A**) The volcano plot of DEGs; (**B**) cluster heat map of DEGs; (**C**) COG annotation classification statistical chart of DEGs; (**D**) GO enrichment analysis of DEGs; (**E**) KEGG classification annotation map; (**F**) KEGG enrichment bubble chart. C and S represent the CK and SPI group respectively.

**Figure 4 jof-10-00500-f004:**
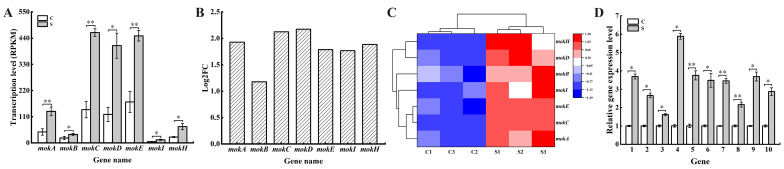
Expression levels of genes related to MK synthesis. (**A**) Transcription level; (**B**) Log2FC; (**C**) expression heat map; (**D**) effects of SPI on *Monascus* gene expression. Genes 1–10 are gene-315043, gene-213838, gene-256816, gene-52089, gene-170694, gene-272292, gene-99680, gene-87025, gene-304829, and gene-194396, respectively. * *p* < 0.05, ** *p* < 0.01.

**Figure 5 jof-10-00500-f005:**
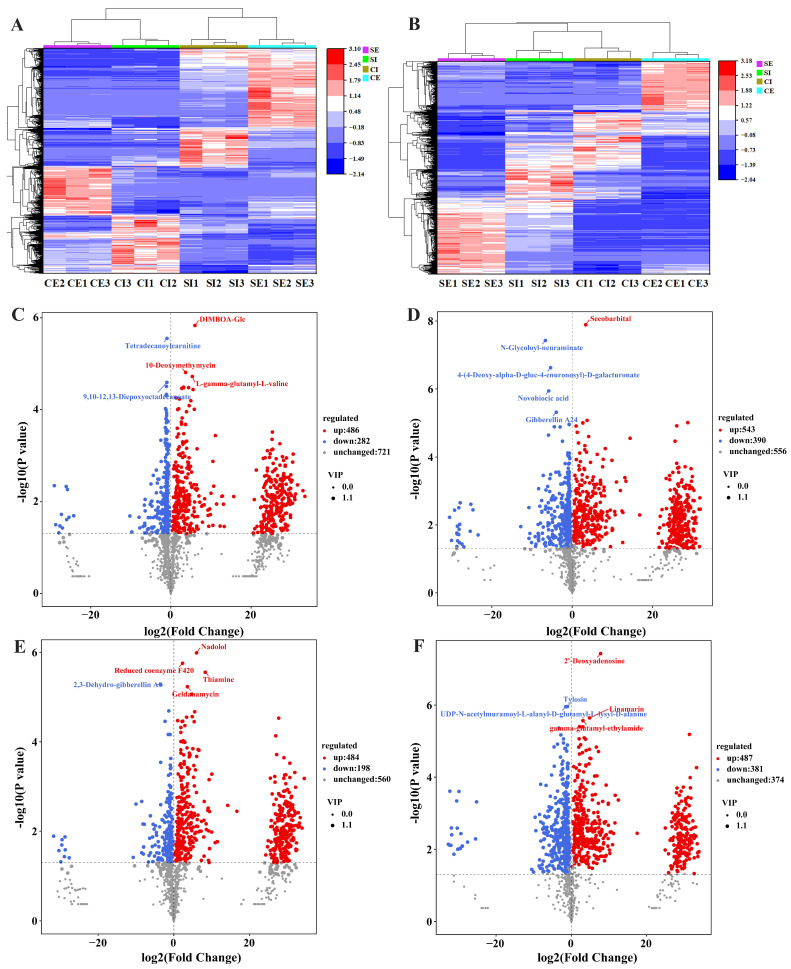
Hierarchical cluster analysis and volcano plot analysis of significant differential metabolites in CK and SPI groups. (**A**) Positive mode; (**B**) negative mode; (**C**) CI vs. SI in positive mode; (**D**) CE vs. SE in positive mode; (**E**) CI vs. SI in negative mode; (**F**) CE vs. SE in negative mode. C and S represent the CK and SPI group, respectively; I and E represent intracellular and extracellular, respectively. The same below.

**Figure 6 jof-10-00500-f006:**
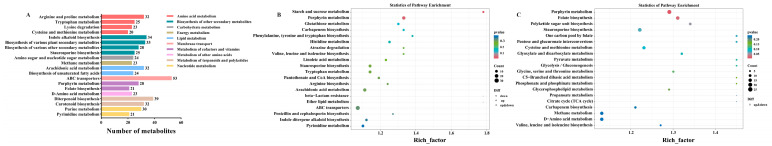
KEGG enrichment map of significant differential metabolic pathway between CK and SPI group. (**A**) KEGG classification annotation map; (**B**) KEGG enrichment bubble chart of intracellular; (**C**) KEGG enrichment bubble chart of extracellular.

**Figure 7 jof-10-00500-f007:**
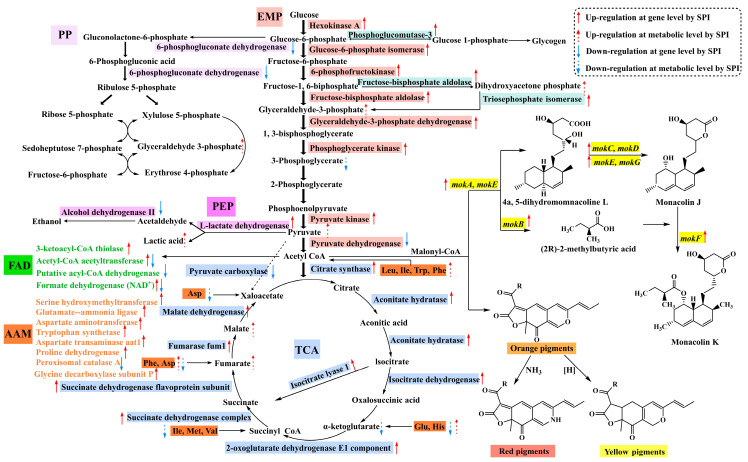
Putative regulatory mechanism of SPI on biosynthesis of MPs and MK. EMP—Embden–Meyerhof pathway; PP—pentose phosphate pathway; PEP—pyruvate metabolism; FAD—fatty acid degradation; AAM—amino acid metabolism; TCA—tricarboxylic acid cycle. Different colors in the figure represent different metabolic pathways, and italics represent genes. Amino acids are represented by abbreviations. Asp: Aspartate, Phe: Phenylalanine, Ile: Isoleucine, Met: Methionine, Val: Valine, Glu: Glutamate, His: Histidine, Leu: leucine, Trp: Tryptophan.

**Table 1 jof-10-00500-t001:** Significant differential metabolites among different groups under positive mode.

Group	ID	Metabolite	Fold Change	log2FC	*p*-Value	VIP	Regulated
CI vs. SI	pos_861	DIMBOA-Glc	64.31	6.01	1.47 × 10^−6^	1.20	up
	pos_6619	Tetradecanoylcarnitine	0.51	−0.97	2.82 × 10^−6^	1.20	down
	pos_2962	10-Deoxymethymycin	12.68	3.66	1.55 × 10^−5^	1.20	up
	pos_1904	L-gamma-glutamyl-L-valine	40.57	5.34	1.90 × 10^−5^	1.20	up
	pos_5448	9,10-12,13-Diepoxyoctadecanoate	0.50	−1.00	2.55 × 10^−5^	1.20	down
	pos_6607	Vanillyl alcohol	0.46	−1.11	3.13 × 10^−5^	1.20	down
	pos_2485	6′-Oxokanamycin C	9.01	3.17	3.24 × 10^−5^	1.20	up
	pos_849	5,6-Epoxytetraene	21.41	4.42	3.32 × 10^−5^	1.20	up
	pos_763	2(alpha-D-Mannosyl)-D-glycerate	8.77	3.13	3.37 × 10^−5^	1.20	up
	pos_2339	1-(2-Furanylmethyl)-1H-pyrrole	6.56	2.71	3.45 × 10^−5^	1.20	up
	pos_987	Pterolactam	46.45	5.54	3.66 × 10^−5^	1.20	up
	pos_6640	5,6-DHET	0.46	−1.11	4.67 × 10^−5^	1.19	down
	pos_6494	α-CEHC	0.48	−1.05	4.91 × 10^−5^	1.19	down
	pos_5923	N,N-dimethyl-Safingol	0.43	−1.22	4.95 × 10^−5^	1.20	down
	pos_2342	Gentioflavine	2.46	1.30	5.57 × 10^−5^	1.20	up
CE vs. SE	pos_3760	Secobarbital	10.38	3.38	1.29 × 10^−8^	1.15	up
	pos_3330	N-Glycoloyl-neuraminate	0.01	−6.66	3.73 × 10^−8^	1.15	down
	pos_513	4-(4-Deoxy-alpha-D-gluc-4-enuronosyl)-D-galacturonate	0.02	−5.37	2.37 × 10^−7^	1.15	down
	pos_4237	Novobiocic acid	0.02	−5.83	1.14 × 10^−6^	1.15	down
	pos_4321	Gibberellin A24	0.06	−3.95	4.86 × 10^−6^	1.15	down
	pos_3233	Hexylglutathione	13.23	3.73	8.45 × 10^−6^	1.15	up
	pos_4667	Priverogenin A	4.67 × 10^8^	28.80	9.76 × 10^−6^	1.15	up
	pos_3242	Methyl succinate	6.36	2.67	9.98 × 10^−6^	1.15	up
	pos_7119	N, N-dimethyl arachidonoyl amine	0.59	−0.75	1.10 × 10^−5^	1.15	down
	pos_3711	5-Oxopentanoate	6.94 × 10^7^	26.05	1.21 × 10^−5^	1.15	up
	pos_2819	Piperidione	1.97	0.98	1.22 × 10^−5^	1.15	up
	pos_544	(11Z,14Z,17Z,20Z,23Z)-Hexacosapentaenoyl-CoA	0.05	−4.44	1.28 × 10^−5^	1.15	down
	pos_2676	3-Methyldioxyindole	0.13	−3.00	1.30 × 10^−5^	1.15	down
	pos_5026	Phe Ala Ile Pro	2.40	1.27	1.88 × 10^−5^	1.15	up
	pos_2066	GDP-valienol	0.02	−5.90	2.26 × 10^−5^	1.15	down

**Table 2 jof-10-00500-t002:** Significant differential metabolites among different groups under negative mode.

Group	ID	Metabolite	Fold Change	log2FC	*p*-Value	VIP	Regulated
CI vs. SI	neg_2428	Nadolol	63.32	5.98	1.03 × 10^−6^	1.20	up
	neg_1245	Reduced coenzyme F420	4.87	2.28	1.76 × 10^−6^	1.20	up
	neg_4963	Thiamine	306.84	8.26	2.79 × 10^−6^	1.20	up
	neg_5554	2,3-Dehydro-gibberellin A9	0.09	−3.45	5.37 × 10^−6^	1.20	down
	neg_3006	Geldanamycin	11.76	3.56	5.86 × 10^−6^	1.20	up
	neg_2431	N-Formylmethionine	26.36	4.72	8.62 × 10^−6^	1.20	up
	neg_1223	Flavine mononucleotide (FMN)	0.39	−1.34	2.04 × 10^−5^	1.20	down
	neg_2901	Triamcinolone Diacetate	44.73	5.48	2.13 × 10^−5^	1.20	up
	neg_5570	4-Allyl-2-methoxyphenol	28.65	4.84	2.74 × 10^−5^	1.20	up
	neg_2806	dTDP-L-olivose	10.17	3.35	2.82 × 10^−5^	1.20	up
	neg_5541	Fumitremorgin C	2.13 × 10^8^	27.66	2.97 × 10^−5^	1.20	up
	neg_6257	5-Aminopentanamide	1.98	0.98	3.37 × 10^−5^	1.20	up
	neg_2782	Phenylacetylglycine	5.05	2.34	3.47 × 10^−5^	1.20	up
	neg_697	Pyroglutamic acid	0.20	−2.33	3.50 × 10^−5^	1.20	down
	neg_5703	Gibberellin A44 diacid	4.46	2.16	3.53 × 10^−5^	1.20	up
CE vs. SE	neg_1676	2′-Deoxyadenosine	218.85	7.77	3.72 × 10^−8^	1.10	up
	neg_6845	Tylosin	0.48	−1.05	1.10 × 10^−6^	1.10	down
	neg_1510	UDP-N-acetylmuramoyl-L-alanyl-D-glutamyl-L-lysyl-D-alanine	0.35	−1.52	1.12 × 10^−6^	1.10	down
	neg_3166	Linamarin	28.44	4.83	2.27 × 10^−6^	1.10	up
	neg_2405	gamma-glutamyl-ethylamide	8.52	3.09	2.70 × 10^−6^	1.10	up
	neg_2784	O-BENZYL-l-SERINE	4.53	2.18	3.94 × 10^−6^	1.10	up
	neg_1104	3′-UMP	8.39	3.07	4.01 × 10^−6^	1.10	up
	neg_1727	3,4-Dihydroxyphthalate	0.10	−3.29	4.33 × 10^−6^	1.10	down
	neg_2997	Ciprofloxacin	2.80 × 10^9^	31.38	6.50 × 10^−6^	1.10	up
	neg_5639	Abacavir	0.14	−2.79	6.72 × 10^−6^	1.10	down
	neg_1598	S (8)-aminomethyldihydrolipoamide	10.73	3.42	8.09 × 10^−6^	1.10	up
	neg_4048	all-trans-4-Hydroxyretinoic acid	4.21	2.07	8.64 × 10^−6^	1.09	up
	neg_1457	C-1027 Chromophore	0.23	−2.09	8.69 × 10^−6^	1.10	down
	neg_3690	3-Ketosucrose	13.89	3.80	8.85 × 10^−6^	1.10	up
	neg_5289	N-Succinyl-L,L-2,6-diaminopimelate	0.24	−2.09	1.13 × 10^−5^	1.10	down

## Data Availability

The raw data supporting the conclusions of this manuscript will be made available by the authors, without undue reservation, to any qualified researcher. The raw RNA-seq data of the present study were deposited into the NCBI database with an accession number of PRJNA1107712.

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
