# Peer review of "Transcriptomic and Metabolomic Analyses of Soybean Protein Isolate on Monascus Pigments and Monacolin K Production"

_jof, 2024, doi:10.3390/jof10070500_

Round 1

Reviewer 1 Report

It seems to me that the main effect that the addition of SPI has on pigment and monacolin production is the formation of acetyl-CoA when amino acids from SPI are utilized by oxidative deamination as carbon sources.  However, this cannot be called a regulatory mechanism. 

It is clear from Figure 1 that M. pilosus forms mainly yellow pigments, and from previous work on this species it is likely that it is a monascuspiloin. There is no mention of this pigment in the paper, neither in the introduction nor in the discussion.

I have no other comments.

Author Response

1.Does the title describe the article's topic with sufficient precision?

It seems to me that the title is not apt. Soybean protein does not have a regulatory effect on monacolin and pigment biosynthesis, but it is a source of precursors for these syntheses.

Response 1:

Thank you for your comments and suggestion. In one respect, soybean protein isolate is a source of precursors for the syntheses of monacolins and pigments, and the utilization of nitrogen source and protease activity during the fermentation process were promoted (Line 285-326). On the other side, under the action of soybean protein isolate, the expression of key genes for monacolins biosynthesis were upregulated ( Line 381-400), and the metabolic pathways were also disturbed (Line 451-460).

Therefore, we embody the regulatory effect of soybean protein isolate in the title.

2. Does the introduction provide a comprehensive yet concise overview about the state of knowledge in the area of research?

I lack detailed information about Monascus pilosus and its pigment production.

Response 2:

Thank you for your comments and suggestion. The experimental strain, M. pilosus MS-1, was isolated from commercial available red mold rice which produces both monacolins and Monascus pigments. The bias production of monacolins or pigments could be regulated by fermentation temperature (Feng Yanli, Identification and fermentation characteristics study of Monascus pilosus MS-1 [D], Huazhong Agricultural University, 2014.) or additives (Shi J, Zhao W, Lu J, Wang W, Yu X, Feng Y*. Insight into Monascus pigments production promoted by glycerol based on physiological and transcriptome analyses[J]. Process Biochem, 2021, 102: 141-149.; Feng Y,Shao Y,Zhou Y,Chen F*.Effects of glycerol on pigments and monacolin K production by the high monacolin K producing but citrinin-free strain, Monascus pilosus MS-1 [J].Eur Food Res Technol,2015,240(3):635-643.).

In the present article, we have added the information about Monascus pilosus and its pigment production in the part of introduction, and the supplementary part was marked in red font (Line 61-72).

“The experimental strain, M. pilosus MS-1, was isolated from commercial available red mold rice which produces both MK and MPs. The bias production of MK or MPs could be regulated by fermentation temperature [18] or additives [19-20]. For example, The MPs yield of M. pilosus MS-1 was 13.78 U/mL when 40 g/L glycerol was added, and the genes related to MPs biosynthesis, ctnR and PKS5, were significantly up-regulated [19]. In addition, when 60 g/L glycerol was added, the MPs production was 3.24 times that of the control group, and the genes MpPKS5, mppR1, MpigI, MpFasA2, and MpFasB2 were significantly up-regulated [20].

Our previous studies indicated that MK production could be selected promoted by soybean flour while that of MPs was inhibited [18]. Then, we found that the key substance affected the productions of MK and MPs was soybean protein isolate (SPI) [21]. On the basis of these, the medium including SPI was optimized for bias production of MK or MPs. ”

3. Is the research design appropriate and are the methods adequately described?

I do not understand whether soybean protein was added into the culture medium already containing monosodium glutamate as nitrogen source oŕ whether monosodium glutamate was alternated with the SPI (2.1)? Further, it is necessary to supplement absorption spectra of both extracellular and intracellular samples containing pigments (2.3) I doubt that Monascus pilosus produces the same pigments as Monascus purpureus and Monascus ruber and therefore it seems strange to measure absorbance at absorbance maxima for yellow, orange and red pigments, which may not correspond to absorbance maxima for M.pilosus. In addition, I do not understand the difference between soluble protein and soluble polypeptide - 2.6.

Response 3:

Thank you for your comments and suggestion.

(1) The soybean protein isolate was added into the culture medium already containing monosodium glutamate as nitrogen source. (2.1)

(2) The Monascus pigments with the color of red, orange or yellow were mixtures which showed non-unified absorption spectra. As we know, there are at least 111 Monascus pigments analogues were identified (Chen W#, Feng Y#, Molnár I*, Chen F*. Nature and nurture: confluence of pathway determinism with metabolic and chemical serendipity diversifies Monascus azaphilone pigments. Natural Product Reports, 2019, 36(4):561-572.). Moreover, the composition of pigments could be varied along with the changes of species and culture conditions. So, besides the total color value at 505 nm, the yellow, orange and red pigments were commonly detected at 390 nm, 460 nm and 505 nm, respectively.

In the present study, the method for determining the Monascus pigments with different colors was based on that of Tong et al. and Shi et al. Tong, A.J.; Lu, J.Q.; Huang, Z.R.; Huang, Q.Z.; Zhang, Y.Y.; Farag, M.A.; Liu, B.; Zhao, C. Comparative transcriptomics discloses the regulatory impact of carbon/nitrogen fermentation on the biosynthesis of Monascus kaoliang pigments. Food Chem. X 2022, 13, 100250. https://dx.doi.org/10.1016/j.fochx.2022.100250;Shi, R.Y.; Gong, P.F.; Liu, Y.T.; Luo, Q.Q.; Chen, W.; Wang, C.T. Linoleic acid functions as a quorum‐sensing molecule in Monascus purpureus–Saccharomyces cerevisiae co‐culture. Yeast 2023, 40, 42-52. https://dx.doi.org/10.1002/yea.3831 (2.3)

(3) The molecular weight of polypeptides is smaller than that of soluble proteins. The determination of polypeptides is carried out after adding trichloroacetic acid to remove acid-insoluble proteins and long-chain peptides. (2.6)

4. English language and style

Moderate editing of English language required

Response 4:

Thank you for your suggestion. We have checked the grammatical errors throughout the manuscript and edited the manuscript by Native speakers for English language, grammar, punctuation, and spelling.

Comment 1: It seems to me that the main effect that the addition of SPI has on pigment and monacolin production is the formation of acetyl-CoA when amino acids from SPI are utilized by oxidative deamination as carbon sources. However, this cannot be called a regulatory mechanism.

Response 1

Thank you for your comments. The soybean protein isolate can be used as a precursor for regulate the synthesis of Monascus pigments and monacolin K. In addition, the transcription level of M. pilosus MS‑1 was significantly affected by soybean protein isolate, and the expression of monacolin K biosynthesis-related genes was significantly promoted by soybean protein isolate. The metabolic level of M. pilosus MS‑1 was significantly affected by soybean protein isolate, and the metabolic pathways were significantly disturbed.

Therefore, we embody the regulatory effect of soybean protein isolate in the title.

Comment 2: It is clear from Figure 1 that M. pilosus forms mainly yellow pigments, and from previous work on this species it is likely that it is a monascuspiloin. There is no mention of this pigment in the paper, neither in the introduction nor in the discussion.

Response 2:

Thank you for your comments. The red, orange, and yellow MPs we measured were all mixtures, and the yellow Monascus pigments were not purified, so we are not sure which one is monascuspiloin. Monascuspiloin is one of the yellow pigments that can be generated by Monascus pilosus. In our future research, the composition of yellow pigments produced by Monascus pilsus MS-1 under the action of soybean isolated protein will be analyzed.

4. Response to Comments on the Quality of English Language

Point 1: Moderate editing of English language required

Response 1: Thank you for your suggestion. We have checked the grammatical errors throughout the manuscript and edited the manuscript by Native speakers for English language, grammar, punctuation, and spelling.

5. Additional clarifications

Thank you for your suggestion. We have checked the funding and edited carefully.

Reviewer 2 Report

soybean protein isolate is too complex

may vary upon sourcing

no clear conclusions could be made with such a complex and variable substrate

no detailed comments

Author Response

1. Does this article provide a relevant contribution to the scientific discussion of this topic?

There are so many transcriptomic and metabolomic analyses conducted on pigment production by Monascus. With very diverse compounds, sometimes complex.... What is/are the active molecule(s) in such complex compounds? Effect of exogenous tannic acid on reduction of biogenic amines and citrinin in monascal soybean and metabolomics analysis Pechyen, C., Uthai, N., Kraboun, K.Food Bioscience, 2024, 59, 103842 Comparative Transcriptomic and Metabolomic Analyses Reveal the Regulatory Effect and Mechanism of Tea Extracts on the Biosynthesis of Monascus Pigments Li, W.-L., Hong, J.-L., Lu, J.-Q., ...Liu, B., Lv, X.-C.Foods, 2022, 11(20), 3159 Authors could write hundred papers, by changing the compound....

Response 1: 

Thank you for your comments. Our previous studies indicated that monacolin K production could be selected promoted by soybean flour while that of Monascus pigments was inhibited. Then, we found that the key substance affected the productions of monacolin K and Monascus pigments was soybean protein isolate. On the basis of these, the medium including soybean protein isolate was optimized for bias production of monacolin K or Monascus pigments.

In the present study, the biosynthesis of yellow Monascus pigments or monacolin K could be preferentially generated under the action of soybean protein isolate. The utilization of nitrogen source, protease activity, transcription level and metabolism level during fermentation were monitored and the regulatory mechanism of soybean protein isolate on Monascus pigments and monacolin K was proposed.

The active molecules in soybean protein isolate that affected Monascus pigments and monacolin K production will be screened based on these results and the results of amino acids during fermentation process (data not shown). This work is carring out.

Feng Y,Shao Y,Zhou Y,Chen F*.Production and optimization of monacolin K by citrinin-free Monascus pilosus MS-1 in solid-state fermentation using non-glutinous rice and soybean flours as substrate [J].Eur Food Res Technol,2014(4), 239:629-636.

Feng Y L,Shao Y C,Zhou Y X,Chen F S*. Monacolin K production by citrinin-free Monascus pilosus MS-1 and fermentation process monitoring [J]. Eng. Life Sci., 2014, 14(5):538-545.

Shi J, Zhang H M, Su Z J, Xu F, Yu X, Feng Y L*. Enhancing monacolin K yield of red yeast rice by adding glucose and substrates from soybean[J]. Food and Fermentation Industries, 2021, 47 (2): 182-187. (In Chinese)

Qin X L, Xie B, Han H L, Zong X L, Hu Y L, Yu X, Feng Y L*. Effect of soybean protein isolate on monacolin K and Monascus pigments production by Monascus spp.[J]. Food and Fermentation Industries, 2023, 1-10. https://doi.org/10.13995/j.cnki.11-1802/ts.037592. (In Chinese)

Comments 1: Soybean protein isolate is too complex

Response 1: 

Thank you for your comments. Our previous study showed that the production of monacolin K was significantly increased under the addition of soybean flours in solid-state fermentation medium. However, adding soybean flours can easily lead to product deterioration due to the presence of fat in the industrial production of red yeast rice. The addition of soybean protein isolate is expected to solve this problem.

The detailed research process was listed as follows.

Our previous studies indicated that monacolin K production could be selected promoted by soybean flour while that of Monascus pigments was inhibited. Then, we found that the key substance affected the productions of monacolin K and Monascus pigments was soybean protein isolate. On the basis of these, the medium including soybean protein isolate was optimized for bias production of monacolin K or Monascus pigments.

In the present study, the biosynthesis of yellow Monascus pigments or MK could be preferentially generated under the action of soybean protein isolate. The utilization of nitrogen source, protease activity, transcription level and metabolism level during fermentation were monitored and the regulatory mechanism of soybean protein isolate on Monascus pigments and monacolin K was proposed.

The active molecules in soybean protein isolate that affected Monascus pigments and monacolin K production will be screened based on these results and the results of amino acids during fermentation process (data not shown). This work is carring out.

Feng Y,Shao Y,Zhou Y,Chen F*.Production and optimization of monacolin K by citrinin-free Monascus pilosus MS-1 in solid-state fermentation using non-glutinous rice and soybean flours as substrate [J].Eur Food Res Technol,2014(4), 239:629-636.

Feng Y L,Shao Y C,Zhou Y X,Chen F S*. Monacolin K production by citrinin-free Monascus pilosus MS-1 and fermentation process monitoring [J]. Eng. Life Sci., 2014, 14(5)538-545.

Shi J, Zhang H M, Su Z J, Xu F, Yu X, Feng Y L*. Enhancing monacolin K yield of red yeast rice by adding glucose and substrates from soybean[J]. Food and Fermentation Industries, 2021, 47 (2): 182-187. (In Chinese)

Qin X L, Xie B, Han H L, Zong X L, Hu Y L, Yu X, Feng Y L*. Effect of soybean protein isolate on monacolin K and Monascus pigments production by Monascus spp.[J]. Food and Fermentation Industries, https://doi.org/10.13995/j.cnki.11-1802/ts.037592. (In Chinese)

Comments 2: May vary upon sourcing

Response 2:

The soybean protein isolate used in the present study is a protein powder with a protein content greater than 90%. The experimental phenomena in this article was comparative analyzed among four Monascus strains in different kinds of media (Qin X L, Xie B, Han H L, Zong X L, Hu Y L, Yu X, Feng Y L*. Effect of soybean protein isolate on monacolin K and Monascus pigments production by Monascus spp.[J]. Food and Fermentation Industries, https://doi.org/10.13995/j.cnki.11-1802/ts.037592. In Chinese).

Comments 3: No clear conclusions could be made with such a complex and variable substrate.

Response 3:

Thank you for your comments.

In the current stage, there are still some unresolved problems. For your question, the synthetic medium was used to eliminate unnecessary interference. What is more, this work is carried out step by step, and the research process was listed in the responses to the previous two questions. We believe that a more clearer conclusion will gradually surfaced as the progress of our research advances.

The active molecules in soybean protein isolate that affected Monascus pigments and monacolin K production will be screened based on these results and the results of amino acids during fermentation process (data not shown). This work is carring out.

4. Response to Comments on the Quality of English Language

Point 1: English language fine. No issues detected

Response 1: Thank you for your suggestion. We have checked the grammatical errors throughout the manuscript and edited the manuscript by Native speakers for English language, grammar, punctuation, and spelling..

5. Additional clarifications

Thank you for your suggestion. We have checked the funding and edited carefully.

Round 2

Reviewer 2 Report

Revision done by authors is fine

Revision done by authors is fine

Author Response

Comments 1: Please change the title and remove "regulatory mechanism" as suggested by the reviewer. I would agree the present study doesn't appear to revel a regulatory mechanism but only changes in transcript levels. I would recommend the new title as "Transcriptomic and Metabolomic Analyses of Soybean Protein Isolate on Monascus Pigments and Monacolin K Production”.

Response 1: Thank you for your comments. We have changed the title to "Transcriptomic and Metabolomic Analyses of Soybean Protein Isolate on Monascus Pigments and Monacolin K Production”.

Comments 2: If this is not already been included, please include this sentence from your response to reviewer somewhere in the methods or results section as suggested by the reviewer. "The soybean protein isolate was added into the culture medium already containing monosodium glutamate as nitrogen source."

Response 2: Thank you for your comments. We have added the content "The soybean protein isolate was added into the culture medium already containing monosodium glutamate as nitrogen source." in the methods and the supplementary part was marked in red font (Line 98-99).

Comments 3: I would tone down using the word regulated in the discussion - maybe use the phrase "changes in" or "altered" instead of regulated.

Response 3: Thank you for your comments. We have used the phrase "changes in" or "effected" instead of regulated in the discussion, and the change part was marked in red font.